# The chronic kidney disease and acute kidney injury involvement in COVID-19 pandemic: A systematic review and meta-analysis

**Ya-Fei Liu[1]\*, Zhe Zhang[1], Xiao-Li Pan[2], Guo-Lan Xing[1], Ying Zhang[1], Zhang-Suo Liu[1], Sheng-Hao Tu[3]**

**1** Department of Nephrology, The First Affiliated Hospital of Zhengzhou University, Zhengzhou, Henan, China, **2** College of Acupuncture and Orthopedics, Hubei University of Chinese Medicine, Wuhan, Hubei, China, **3** Institute of Integrated Traditional Chinese and Western Medicine, Tongji Hospital, Tongji Medical College, Huazhong University of Science and Technology, Wuhan, Hubei, China

\* yafeiliutjh@gmail.com

**Data Availability Statement:** All relevant data are in the paper and its Supporting Information files.

**Funding:** YFL received a National Natural Science Foundation of China (81701601). http://www.nsfc.

## Abstract

### Background

Currently, the SARS-CoV-2 promptly spread across China and around the world. However, there are controversies about whether preexisting chronic kidney disease (CKD) and acute kidney injury complication (AKI) are involved in the COVID-19 pandemic.

### Measurements

Studies reported the kidney outcomes in different severity of COVID-19 were included in this study. Standardized mean differences or odds ratios were calculated by employing Review Manager meta-analysis software.

### Results

Thirty-six trials were included in this systematic review with a total of 6395 COVID-19 patients. The overall effects indicated that preexisting CKD (OR = 3.28), complication of AKI (OR = 11.02), serum creatinine (SMD = 0.68), abnormal serum creatinine (OR = 4.86), blood urea nitrogen (SMD = 1.95), abnormal blood urea nitrogen (OR = 6.53), received continuous renal replacement therapy (CRRT) (OR = 23.63) were significantly increased in severe group than that in nonsevere group. Additionally, the complication of AKI (OR = 13.92) and blood urea nitrogen (SMD = 1.18) were remarkably elevated in the critical group than that in the severe group.

### Conclusions

CKD and AKI are susceptible to occur in patients with severe COVID-19. CRRT is applied frequently in severe COVID-19 patients than that in nonsevere COVID-19 patients. The risk of AKI is higher in the critical group than that in the severe group.

gov.cn/ The funders had no role in study design, data collection and analysis, decision to publish, or preparation of the manuscript.

**Competing interests:** The authors have declared that no competing interests exist.

## Introduction

In December, 2019, 41 hospitalized patients had been identified as having laboratory-confirmed 2019-novel coronavirus (2019-nCoV)-related infection in Wuhan, Hubei, China [1]. The 2019-nCoV was discovered by whole-genome sequencing, direct PCR, and culture [2]. As of October 20, 2020, 40,657,387 laboratory-confirmed cases had been documented worldwide. The single-cell RNA sequencing datasets indicated that angiotensin converting enzyme 2 (ACE2) was mainly expressed in lung type II alveolar cells, colon colonocytes, ileum ECs, and proximal tubule cells [3, 4]. In addition, SARS-CoV-2 was detected in urine, blood, anal swab, and oropharyngeal swab from nine patients with COVID-19 who were retested by qRT-PCR [5]. Moreover, the single-cell transcriptome analysis identified that ACE2 and transmembrane protease serine 2 (TMPRSS2) genes are highly coexpressed in podocytes and proximal tubule cells [6].

A prospective cohort study of 701 patients with COVID-19 showed that the prevalence of increased serum creatinine, elevated blood urea nitrogen, and estimated glomerular filtration (eGFR) < 60 ml/min/1.73m$^2$ were 14.4, 13.1 and 13.1%, respectively [7]. Meanwhile, acute kidney injury (AKI) occurred in 5.1% COVID-19 patients [7]. A meta-analysis indicated that chronic kidney disease (CKD) was consistent with severe COVID-19 [8]. Furthermore, Chinese professionals point out that AKI cannot be ignored in the diagnosis and treatment of COVID-19 [9]. However, one study demonstrated that SARS-CoV-2 infection does not result in AKI or aggravate CKD in COVID-19 patients [10]. Therefore, it is stringent to perform a systematic review to verify whether preexisting CKD is a risk factor for severe COVID-19 and AKI is a complication for severe COVID-19.

## Methods

The Preferred Reporting Items for Systematic Review and Meta-Analyses (PRISMA) statement was employed to design and report the study.

### Search strategy

The following English databases were retrieved to confirm trials: PubMed and medRxiv.gov. In addition, the Chinese databases, such as the CNKI Database and WanFang Database were searched. All the databases were searched from their available dates of inception to the latest issue (April 13, 2020).

For the English databases, free text terms were employed, such as "SARS-CoV-2" or "2019-nCoV" or "COVID-19" or "novel coronavirus pneumonia" and "kidney" or "renal". For the Chinese databases, free text terms were applied for "xin xing guan zhuang bing du" (which means SARS-CoV-2 in Chinese) or "xin xing guan Zhuang bing du fei yan" (which means COVID-19 in Chinese). The reference lists of relevant publications were also searched to identify extra studies.

### Selection criteria

Inclusion criteria: (1) study type: case series, cohort study, or prospective study that reported the connection between kidney involvement and COVID-19 irrespective of publication status or language; (2) subjects: adult patients diagnosed with COVID-19; (3) outcomes: COVID-19 patients with preexisting CKD, AKI, serum creatinine, blood urea nitrogen, or received continuous renal replacement therapy (CRRT).

Case reports, reviews, meta-analyses, or studies regarding receiving renal replacement therapy regularly were excluded. For repeat studies whose preprints were published in medRxiv.

gov, the final version studies were included. The literatures were selected by two reviewers (YF Liu and Z Zhang) independently.

## Data extraction and management

The data were extracted by two independent reviewers (XL Pan and GL Xing), and contradictions were resolved by consensus or were judged by another author. The data presented as median and interquartile range were transformed into mean and standard deviation according to the formula below (http://www.math.hkbu.edu.hk/~tongt/papers/median2mean.html).

The studies' quality was evaluated according to the Newcastle-Ottawa scale (NOS). The quality of the included studies was estimated independently by two investigators (SH Tu and Y Zhang). The NOS scores $\geq$ 6 were considered high quality studies. The authors were connected to clarify the vagueness or absence of the data, and related data were extracted by consensus if the authors were unavailable.

## Data synthesis and analysis

The Review Manager software 5.3 was applied to analyze the indicators. The standardized mean differences (SMDs) and 95% confidence intervals (CIs) were calculated for continuous data. The odds ratios (ORs) and 95% CIs were calculated for dichotomous data. Heterogeneity was estimated through the chi-square test and Higgins $I^2$ test. A fixed-effect model was applied if the studies were sufficiently alike ($P > 0.10$); otherwise, a random-effects model was employed. A Z score was calculated to detect the overall effect, with a significance set at $P < 0.05$. Publication bias was evaluated by funnel plot if the number of included studies $> 10$.

Two subgroup analyses were performed to diminish the clinical heterogeneity according to the disease severity: severe group compared with nonsevere group, critical group compared with severe group.

## Results

### Study selection

After filtering, COVID-19 patients were divided into severe and nonsevere groups in 28 studies [1, 12–37] (Fig 1). In 8 studies, the COVID-19 patients were classified as more than two groups according to the disease severity, which included severe and critical COVID-19 patients [38–45]. Finally, 36 studies were included in the study. The characteristics of the studies were indicated in Table 1. Together, 6395 COVID-19 patients were included in the study.

### Study descriptions

All the included studies were performed in China. Five studies were published in Chinese [27–29, 44, 45], and the others were published in English. Eight studies were conducted as multicenter trials [11, 12, 16, 20–22, 26, 42]. The patients with COVID-19 divided into ICU care and no ICU care in four studies were included in the severe and nonsevere subgroup [1, 33, 34, 36]. The COVID-19 patients divided into ARDS (acute respiratory distress syndrome) and non-ARDS in one study were also included in the severe and nonsevere subgroup [35]. The two studies combined the data of severe and critical patients together, and they were also included in the severe and nonsevere subgroup [27, 28].

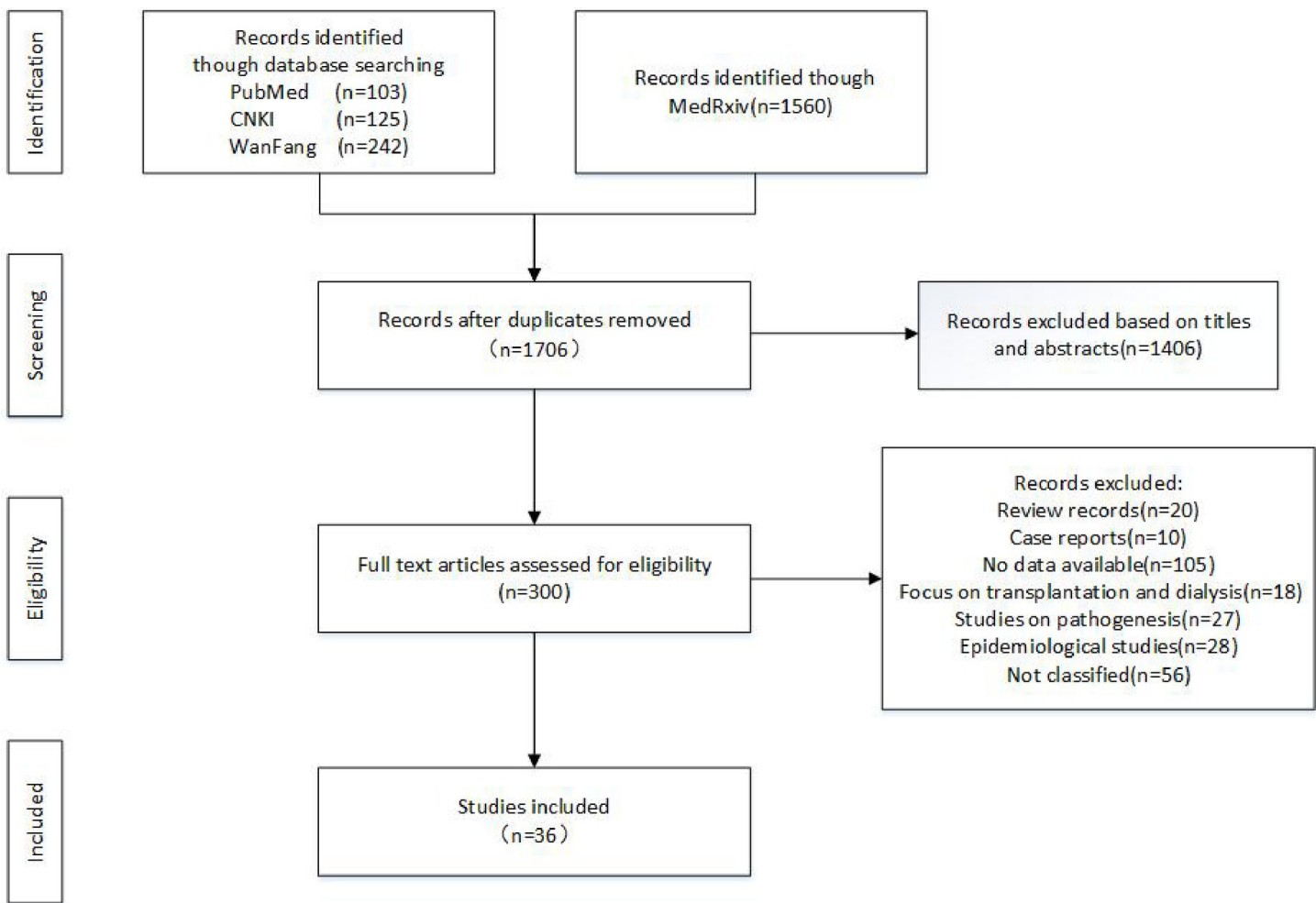

**Fig 1. Flowchart of study selection.**

## Quality of the included studies

As indicated in Table 1, the majority of the included studies were poor quality except four studies whose NOS scores are 6 [1, 13, 28, 40]. The NOS scores of the other 31 trials are 5, and only one trial's NOS scores were 4.

## Publication bias

As the included studies of three outcomes > 10 in severe and nonsevere subgroup, funnel plots were conducted to estimate the publication bias (Fig 2). The funnel plot of preexisting CKD was symmetrical, suggesting that there was no publication bias. However, the funnel plot of serum creatinine and blood urea nitrogen were a little asymmetrical, implying that the publication bias existed to some extent.

## Effects of outcomes

**Severe group compared with nonsevere group.** *Propotion of preexisting CKD*. The number of participants ranged from 21 to 1099. There was no statistical heterogeneity between the studies

**Table 1. Clinical and demographic characteristics of the patients with COVID-19.**

| Study | Age | Male No. (%) | Female No. (%) | Type of study | Region | Outcomes | NOS score |
|---|---|---|---|---|---|---|---|
| Xu S 2020 | ▲ | 193 (54.4) | 162 (45.6) | Retrospective study | Wuhan, Fuyang, China | ②③⑦ | 5 |
| Guan W 2020 | 47 (35.0–58.0) △ | 637 (58.1) | 459 (41.9) | Cohort study | 30 provinces, China | ①⑤⑥⑦ | 5 |
| Yang JK 2020 | 61 (52–67) △ | 34 (49.3) | 35 (50.7) | Retrospective cohort study | Wuhan, China | ②③ | 5 |
| Li Z 2020 | 57 (46–67) △ | 95 (49) | 98 (51) | Retrospective study | Wuhan, Huangshi, Chongqing, China | ①②③④⑤⑥⑦ | 5 |
| Chen G 2020 | 56.3 ± 14.3 ▽ | 17 (81.0) | 4 (19.0) | Retrospective study | Wuhan, China | ②③ | 6 |
| Hui H 2020 | NA | 19 (46.3) | 22 (53.7) | Retrospective study | Beijing, China | ③ | 5 |
| Zhao W 2020 | 52 ± 20▽ | 34 (44.2) | 43 (55.8) | Retrospective cohort study | Beijing, China | ①③⑤⑦ | 5 |
| Wang YF 2020 | ▲ | 48 (43.6) | 62 (56.4) | Retrospective study | Wuhan, China | ②③ | 5 |
| Xu Y 2020 | 57 (43–69) △ | 35 (50.7) | 34 (49.3) | Retrospective case series study | Wuhan, Shanghai, Beijing, China | ③ | 5 |
| Cao WL 2020 | ▲ | 60 (46.9) | 68 (53.1) | Retrospective study | Xiangyang, China | ②③ | 5 |
| Zhang GQ 2020 | 55.0 (39.0–66.5) △ | 108 (48.9) | 113 (51.1) | Retrospective case series study | Wuhan, China | ①②③⑥⑦ | 5 |
| Liu YL 2020 | 55 (43–66) △ | 59 (54.1) | 50 (45.9) | Retrospective study | Wuhan, China | ①②③ | 5 |
| Cao M 2020 | 50.1 ± 16.3 ▽ | 101 (51.0) | 97 (49.0) | Cohort study | Shanghai, China | ②③④⑤ | 5 |
| Liu L 2020 | 45 (34–51) △ | 32 (62.7) | 19 (37.3) | Retrospective case series study | Chongqing, China | ②③ | 5 |
| Yan SJ 2020 | 51 (36–62) △ | 81 (48.2) | 87 (51.8) | Retrospective study | Hainan province, China | ①②③④⑤⑦ | 4 |
| Chen XH 2020 | 64.6 ± 18.1▽ | 37 (77.1) | 11 (22.9) | NA | Wuhan, China | ②③ | 6 |
| Qian GQ 2020 | 50 (36.5–57) △ | 37 (40.66) | 54 (59.34) | Retrospective case series | Zhejiang province, China | ②③ | 5 |
| Qi D 2020 | 48.0 (35.0–65.0) △ | 149 (55.8) | 118 (44.2) | Retrospective study | Chongqing, China | ⑤ | 5 |
| Liu JY 2020 | 40 (1–86) ▾ | 31 (50.8) | 30 (49.2) | Prospective study | Beijing, China | ②③ | 5 |
| Xiang JL 2020 | NA | 15 (53.6) | 13 (56.4) | NA | Zunyi, China | ① | 5 |
| Huang H 2020 | 44.87 ± 18.55▽ | 63 (50.4) | 62 (49.6) | Retrospective study | Guangzhou, China | ③ | 5 |
| Hu L 2020 | 61 (23–91) ▾ | 166 (51.4) | 157 (48.6) | Retrospective study | Wuhan, China | ①④⑤⑦ | 5 |
| Feng ZC 2020 | 47 (36–58) ▾ | 284 (50.4) | 280 (49.6) | Retrospective cohort study | Hunan province, China | ①②③ | 5 |
| Zhou HF 2020 | 47.0 (35.0–61.0) △ | 72 (40.4) | 106 (59.6) | Retrospective study | Wuhan, China | ②③④⑤ | 5 |
| Fang XW 2020 | 45.1 ± 16.6 ▽ | 45 (57) | 34 (43) | Retrospective study | Anhui province, China | ①②③ | 5 |
| Gao W 2020 | 51.7 ± 18.6 ▽ | 43 (47) | 47 (53) | Retrospective study | Beijing, China | ③ | 5 |
| Xiang TX 2020 | 42.9 (18–78) ▾ | 33 (67.3) | 16 (32.7) | Retrospective study | Jiangxi province, China | ②③ | 6 |
| Cheng KB 2020 | 51 (43,60) △ | 244 (52.7) | 219 (47.3) | Retrospective study | Wuhan, China | ①②③ | 5 |
| Xiong J 2020 | 53.0 ± 16.9 ▽ | 41 (45.6) | 49 (54.4) | Retrospective study | Wuhan, China | ①②③⑦ | 5 |
| Huang CL 2020 | 49.0 (41.0–58.0) △ | 30 (73) | 11 (27) | Cohort study | Wuhan, China | ③⑥⑦ | 6 |
| Wan SX 2020 | 47 (36–55) △ | 72 (53.3) | 63 (46.7) | NA | Chongqing, China | ③⑥⑦ | 5 |
| Wang DW 2020 | 56 (42–68) △ | 75 (54.3) | 63 (45.7) | Retrospective, case series | Wuhan, China | ①②③⑥⑦ | 5 |
| Zhang JJ 2020 | 57 (25–87) ▾ | 71 (50.7) | 69 (49.3) | NA | Wuhan, China | ① | 5 |
| Feng Y 2020 | 53 (40–64) △ | 271 (56.9) | 205 (43.1) | Retrospective study | Wuhan, Shanghai, Anhui, China | ①②③ | 5 |
| Fan LC 2020 | 46.8□ | 30 (54.5) | 25 (45.5) | Retrospective, study | Shenyang, China | ③ | 5 |
| Shen B 2020 | 47.7 ± 13.9 ▽ | 31 (67.4) | 15 (32.6) | Cohort study | Zhejiang province, China | ① | 5 |

Note: Age was expressed as different levels▲ or median and interquartile range△ or median and range▾ or mean ± standard deviation▽ or mean□. NA: not available. ① Chronic kidney disease; ② Blood urea nitrogen; ③ Serum creatinine; ④ Abnormal blood urea nitrogen; ⑤ Abnormal serum creatinine; ⑥ Continuous renal replacement therapy; ⑦ Acute kidney injury.

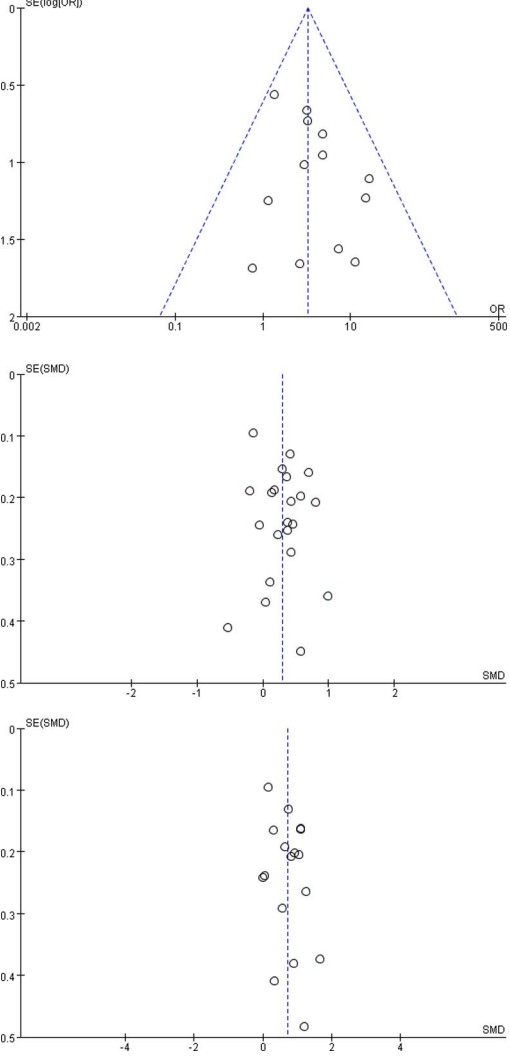

**Fig 2. The funnel plots.** CKD: chronic kidney disease.

($P$ = 0.72). Thirteen studies mentioned preexisting CKD (including 3325 subjects), and pooled results indicated that propotion of preexisting CKD was significantly increased in the severe group than that in the nonsevere group ($P < 0.00001$, OR = 3.28, 95% CI: 2.0 to 5.37) (Fig 3).

*Complication of AKI.* The number of participants ranged from 41 to 1099. There was no statistical heterogeneity between the studies ($P$ = 0.12). Eight studies reported a complication of AKI (involving 2072 subjects), and the complication of AKI was significantly elevated in the severe group than that in the nonsevere group ($P < 0.00001$, OR = 11.02, 95% CI: 6.54 to 18.57) (Fig 3).

*Serum creatinine.* The number of participants ranged from 21 to 564. There was statistical heterogeneity between the studies ($P < 0.00001$). Twenty-two studies measured serum creatinine (including 3261 subjects), and pooled results showed that serum creatinine was remarkably increased in the severe group than that in the nonsevere group ($P$ = 0.0005, SMD = 0.68, 95% CI: 0.3 to 1.06) (Fig 3).

*Abnormal serum creatinine.* The number of participants ranged from 77 to 752. There was no statistical heterogeneity between the studies ($P$ = 0.51). Seven studies documented abnormal

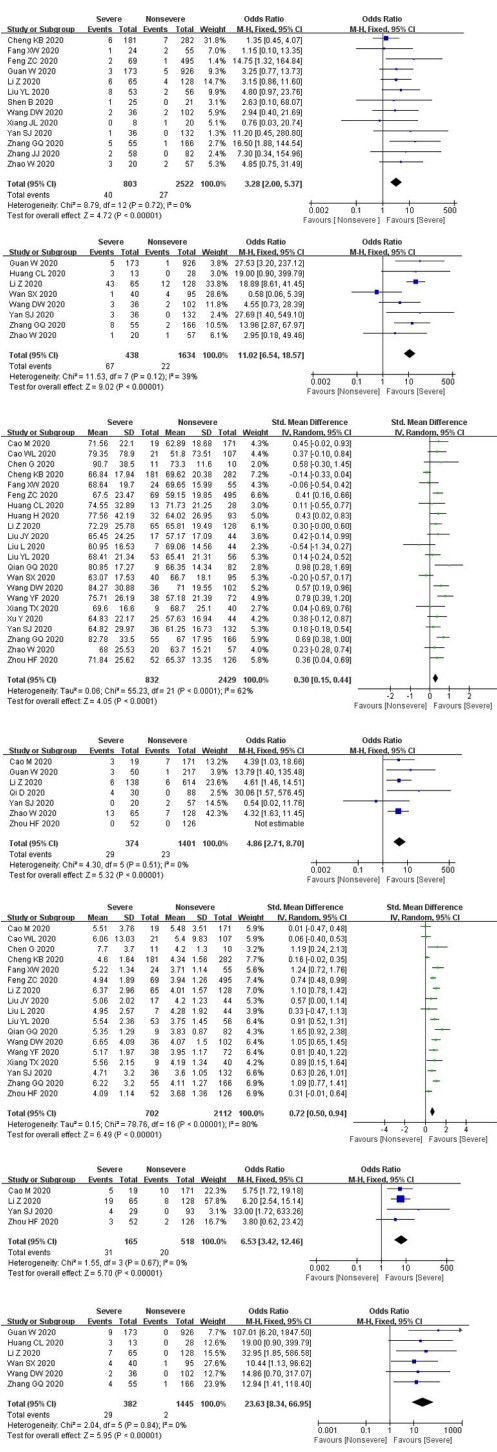

**Fig 3. Forest plots for the severe and nonsevere subgroup.** CKD: chronic kidney disease; AKI: acute kidney injury; CRRT: continuous renal replacement therapy.

serum creatinine (including 1775 subjects), and the severe group had a higher ratio of abnormal serum creatinine than the nonsevere group ($P < 0.00001$, OR = 4.86, 95% CI: 2.71 to 8.7) (Fig 3).

*Blood urea nitrogen.* The number of participants ranged from 21 to 564. There was statistical heterogeneity between the studies ($P < 0.00001$). Seventeen studies assayed blood urea

nitrogen (including 2814 subjects), and pooled results revealed that blood urea nitrogen was significantly elevated in the severe group than that in the nonsevere group ($P < 0.00001$, SMD = 1.95, 95% CI: 1.23 to 2.66) (Fig 3).

*Abnormal blood urea nitrogen.* The number of participants ranged from 122 to 193. There was no statistical heterogeneity between the studies ($P = 0.67$). Four studies documented abnormal blood urea nitrogen (involving 683 subjects), and the severe group had a higher ratio of abnormal blood urea nitrogen than the nonsevere group ($P < 0.00001$, OR = 6.53, 95% CI: 3.42 to 12.46) (Fig 3).

*Receiving CRRT.* The number of participants ranged from 41 to 1099. There was no statistical heterogeneity between the studies ($P = 0.84$). Six studies reported receiving CRRT (including 1827 subjects), and severe group had higher ratio of receiving CRRT than nonsevere group ($P < 0.00001$, OR = 23.63, 95% CI: 8.34 to 66.95) (Fig 3).

**Critical group compared with severe group.** *Propotion of preexisting CKD.* The number of participants ranged from 50 to 166. There was no statistical heterogeneity between the studies ($P = 0.88$). Three studies reported preexisting CKD (including 346 subjects), and the pooled results indicated that there was no significant difference between the two groups ($P = 0.77$, OR = 1.3, 95% CI: 0.23 to 7.16) (Fig 4).

*Complication of AKI.* The number of participants ranged from 50 to 172. There was no statistical heterogeneity between the studies ($P = 0.37$). Three studies documented the complication of AKI (involving 354 subjects), and the complication of AKI was significantly elevated in the critical group than that in the severe group ($P < 0.00001$, OR = 13.92, 95% CI: 4.69 to 41.26) (Fig 4).

*Serum creatinine.* The number of participants ranged from 27 to 132. There was statistical heterogeneity between the studies ($P < 0.00001$). Seven studies determined serum creatinine (including 480 subjects), and pooled results indicated that there was no significant difference between the two groups ($P = 0.13$, SMD = 0.43, 95% CI: -0.13 to 0.99) (Fig 4).

*Blood urea nitrogen.* The number of participants ranged from 27 to 132. There was statistical heterogeneity between the studies ($P < 0.00001$). Five studies measured blood urea nitrogen (including 379 subjects), and pooled results showed that blood urea nitrogen is strikingly

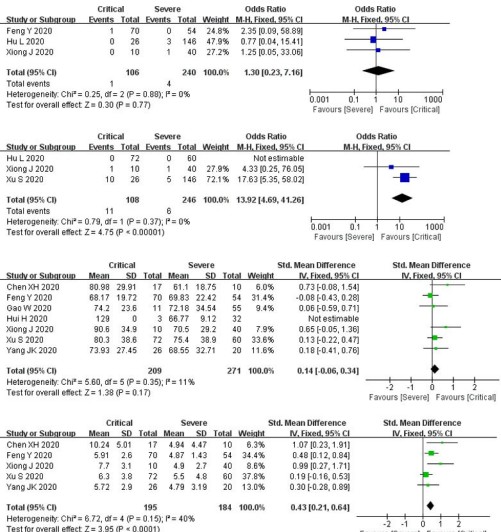

**Fig 4. Forest plots for the critical and severe subgroup.** CKD: chronic kidney disease; AKI: acute kidney injury.

increased in the critical group than that in the severe group (*P* = 0.002, SMD = 1.18, 95% CI: 0.44 to 1.91) (Fig 4).

Only one study reported abnormal blood urea nitrogen and abnormal serum creatinine, so the result was not pooled. CRRT was not documented in this subgroup.

## Discussion

To our knowledge, this is one of the most comprehensive research which illustrates the kidney involvement in COVID-19 pandemic. Compared with the nonsevere group, propotion of pre-existing CKD (OR = 3.28) and complication of AKI (OR = 11.02) were significantly elevated in the severe group. Meanwhile, AKI occurred in 4.3% COVID-19 patients, and 21.1% were severe COVID-19 patients. The complication of AKI is lower than 5.1% which was reported in a previous cohort study, of which 42.7% were severely ill [7]. The different disease severity and sample size may contribute to the difference. The propotion of preexisting CKD is similar to the previous meta-analysis (including 1389 participants), which showed that CKD was significantly associated with severe COVID-19 (OR = 3.03), among which 19.7% were defined as severe COVID-19 [8].

Furthermore, serum creatinine, ratio of abnormal serum creatinine, blood urea nitrogen, ratio of abnormal blood urea nitrogen were remarkably elevated in the severe group than that in the nonsevere group, with SMD = 0.68, OR = 4.86, SMD = 1.95, OR = 6.53, respectively. The OR regarding abnormal serum creatinine and abnormal blood urea nitrogen were lower than that in the severe and nonsevere groups regarding AKI. The AKI also occurred in normal serum creatinine and blood urea nitrogen, which could lead to the higher OR of AKI.

CRRT is recommended in severe or critical patients because it reduces cytokine damage, removes endotoxin, and improves cardiac function and kidney function [46]. Hence, the severe group had a higher ratio of receiving CRRT than the nonsevere group (OR = 23.63).

The critical group had a higher ratio of AKI than that in the severe group (OR = 13.92), which is higher than the severe and nonsevere subgroup (OR = 11.02). Furthermore, the blood urea nitrogen was increased in the critical group than that in the severe group (SMD = 1.18). However, there is no significant difference between the two groups regarding propotion of preexisting CKD. Owing to immunodepression of CKD, the cytokine storm is weakened and CKD patients are not vulnerable to critical COVID-19, which could explain the result. Meanwhile, there was no significant increase in serum creatinine in the critical group. We speculated that the CRRT was frequently applied in the critical group, so serum creatinine was not notably elevated in the critical group. In addition, AKI is diagnosed not only by abnormal serum creatinine but also the change of serum creatinine and urine volume.

There are lots of reasons that give rise to the kidney impairment in COVID-19. First, SARS-CoV-2 viral loads induce cytotoxicity of renal resident cells. The endothelial cell swelling, renal tubular epithelial cell swelling, vacuolar degeneration, and drop were observed with light microscope of renal biopsy in three COVID-19 cases [47]. Second, fever, vomit, diarrhea, and shock are likely to cause kidney hypoperfusion. Third, serum SARS-CoV-2 viral load is strongly associated with cytokine storm [40]. A renal transplanted patient with COVID-19 showed mild disease because immunosuppression of renal transplantation can "be protective" [48]. Fourth, the nephrotoxicity of some drugs. Fifth, organ crosstalk such as cardiorenal syndrome, hypoxia, and rhabdomyolysis also could cause kidney impairment [46]. Sixth, mechanical ventilation is able to result in AKI [49]. Seventh, elderly patients with comorbidities such as hypertension, diabetes, chronic cardiology disease, or chronic liver disease are susceptible to trigger secondary kidney diseases which may not be confirmed before admission.

Several limitations of the included studies should be concerned. First, all the included studies did not stage CKD. Different CKD stages could have different risks for COVID-19 and have different prognosis. Second, only one study [7] reported the AKI stage, which reflects the AKI severity, progression, and prognosis. AKI occurred primarily or AKI on CKD is unknown for the included studies. Third, all the included studies did not qualify the 24h total urine protein. Fourth, all the included studies did not measure the 24h urine volume or the change of urine volume. Only two studies reported the urine routine [12, 36]. Finally, the outcomes of AKI were unclear.

Several limitations of this study should be noticed. First, all the included trials were conducted in Chinese populations, which inferred a high risk of selection bias. Second, most of the included studies were of poor quality except four studies. Third, the heterogeneity was significant in terms of serum creatinine and blood urea nitrogen. The different disease severity, sample size, detection methods, and reference range were responsible for the heterogeneity. Fourth, children or teenagers COVID-19 patients were probably included in this study because the included studies did not clearly restrict to adult COVID-19 patients. Fifth, the quality of most of the included studies was low, which may weaken the evidence level. Finally, the most important criterion eGFR was reported in only three studies [27, 34, 44].

## Conclusion

In conclusion, this systematic review suggests that preexisting CKD and AKI complication are involved in severe COVID-19. Severe COVID-19 is susceptible to occur in preexisting CKD patients than nonsevere COVID-19. With the severity of COVID-19 increasing, the risk of AKI elevated strikingly. CRRT is applied more frequently in the severe group than that in the nonsevere group. More attention should be paid to monitoring kidney function to avoid end stage renal disease and decease in COVID-19 pandemic.

## Supporting information

**S1 Data.**
(XLSX)

**S1 Checklist. PRISMA checklist.**
(DOC)

**S1 File. NOS score.**
(XLS)

## Acknowledgments

We would like to acknowledge medical staffs who fight in the first line of COVID-19 pandemic.

## Author Contributions

**Data curation:** Zhe Zhang, Xiao-Li Pan, Guo-Lan Xing.

**Formal analysis:** Zhe Zhang.

**Investigation:** Ying Zhang.

**Supervision:** Ya-Fei Liu, Ying Zhang, Sheng-Hao Tu.

**Writing – original draft:** Ya-Fei Liu.

**Writing – review & editing:** Xiao-Li Pan, Zhang-Suo Liu, Sheng-Hao Tu.

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
