## [Decision Letter · Decision Letter 0]

3 Sep 2020

PONE-D-20-22031

The Chronic Kidney Disease and Acute Kidney Injury Involvement in COVID-19 Pandemic: A Systematic Review and Meta-analysis

PLOS ONE

Dear Dr. Liu,

Thank you for submitting your manuscript to PLOS ONE. After careful consideration, we feel that it has merit but does not fully meet PLOS ONE’s publication criteria as it currently stands. Therefore, we invite you to submit a revised version of the manuscript that addresses the points raised during the review process.

We look forward to receiving your revised manuscript.

Kind regards,

Tatsuo Shimosawa, M.D., Ph.D.

Academic Editor

PLOS ONE

Journal Requirements:

2.We note that you have indicated that data from this study are available upon request. PLOS only allows data to be available upon request if there are legal or ethical restrictions on sharing data publicly. For information on unacceptable data access restrictions, please see http://journals.plos.org/plosone/s/data-availability#loc-unacceptable-data-access-restrictions.

3.PLOS requires an ORCID iD for the corresponding author in Editorial Manager on papers submitted after December 6th, 2016. Please ensure that you have an ORCID iD and that it is validated in Editorial Manager. To do this, go to ‘Update my Information’ (in the upper left-hand corner of the main menu), and click on the Fetch/Validate link next to the ORCID field. This will take you to the ORCID site and allow you to create a new iD or authenticate a pre-existing iD in Editorial Manager. Please see the following video for instructions on linking an ORCID iD to your Editorial Manager account: https://www.youtube.com/watch?v=_xcclfuvtxQ

<h1>** **</h1>

Reviewers' comments:

Reviewer's Responses to Questions

**Comments to the Author**

1. Is the manuscript technically sound, and do the data support the conclusions?

Reviewer #1: Yes

Reviewer #2: No

2. Has the statistical analysis been performed appropriately and rigorously? 

Reviewer #1: I Don't Know

Reviewer #2: I Don't Know

3. Have the authors made all data underlying the findings in their manuscript fully available?

Reviewer #1: Yes

Reviewer #2: Yes

4. Is the manuscript presented in an intelligible fashion and written in standard English?

Reviewer #1: Yes

Reviewer #2: No

5. Review Comments to the Author

Reviewer #1: Authors conducted a systematic review & meta-analysis of risk of kidney involvement according the severity of COVID19. Although the objective of the manuscript is interesting enough and deserves publication, the current version of the manuscript is incomplete in terms of quality according to PRISMA guideline and needs revisions according to the comments below.

* First paragraph of the Introduction should be more concise and to the point.

* "to verify whether kidney diseases are involved in COVID19" is very vague. Objective of the study should be more clearly defined (authors seems to only assess increasing prevalence of CKD or AKi according to the severity of COVID19).

* Authors should provide SR registration number as in PROSPERO.

* In selecting the studies, AKI or CKD should be clearly defined, which seems vague in this study.

* Studies selected are only from China, but those from other area should be included for applicability of the study results.

* Authors should provide at least one literature search formula, so that readers/reviewers can reproduce the study results.

* Risk of Bias table should be presented. For NOS score, only the total score is not enough but needs to present which item acquired the score in each selected study.

* Last paragraph of the Discussion should not be the recommendation. Recommendation should be derived from the current study, but it did not provide any information for the recommended items.

* Conclusion should be confined to the one derived from the results gained in this study. Need for CRRT cannot be concluded from the study.

Reviewer #2: This review article evaluated involvement of CKD and AKI in COVID-19 patients. Due to the high impact of COVID-19 on public healthcare all over the world, I believe there is useful information in this manuscript. However, there are many inaccurate wordings in the current manuscript, which make it difficult to understand the appropriate conclusions from this study. Overall, I believe what the authors found were 1. CKD is a risk factor of severe COVID-19, and 2. Higher complication rates of AKI were found in severe COVID-19 patients when compared to non-severe cases. This is something already reported, but additional publications should be welcomed by researchers. Followings are my major comments.

1. The definition of severe or non-severe COVID-19 cases is unclear. Because of this, it was not possible to evaluate the manuscript appropriately. Does the severe case mean patients who required RRT, intubation, or something else?

2. In the introduction, the number of COVID patient is outdated. Please update.

3. At the end of introduction, authors said “to verify whether kidney diseases are involved in COVID-19”. This is an unclear statement. If the purpose of this study is to determine whether preexisting CKD is a risk factor for severe COVID19, then authors should explain so. If the purpose was to evaluate whether AKI is a complication in severe COVID19 cases, that should be also explained.

4. In study description, authors explained “All the included studies were performed in China”. What is the reason why studies from outside China were excluded?

5. Due to unclear wording, it is difficult to understand the results. For example, in Comorbidity of CKD, authors said “the comorbidity of CKD was significantly increased in the severe group than that in the nonsevere groups.”. This sounds like the CKD was developed more frequently in severe group after COVID19 infection. But, I believe what authors meant was the patients with CKD were more likely to develop severe COVID19. The same is true to serum creatinine, blood urea nitrogen, and etc.

6. In the result section, “serum creatinine” and “blood urea nitrogen” were repeated twice. This is confusing and needs to be corrected.

7. In discussion, authors claimed this is the first research to evaluate kidney involvement in COVID-19 patients. There are many papers published. This is probably overclaim and needs to be revised.

6. PLOS authors have the option to publish the peer review history of their article (what does this mean?). If published, this will include your full peer review and any attached files.

Reviewer #1: No

Reviewer #2: No

---

## [Author Response · Author response to Decision Letter 0]

25 Nov 2020

Reviewer #1: Authors conducted a systematic review & meta-analysis of risk of kidney involvement according the severity of COVID19. Although the objective of the manuscript is interesting enough and deserves publication, the current version of the manuscript is incomplete in terms of quality according to PRISMA guideline and needs revisions according to the comments below.

1. First paragraph of the Introduction should be more concise and to the point.

Response: According to your suggestion, we have revised the first paragraph of the Introduction.

2. "to verify whether kidney diseases are involved in COVID19" is very vague. Objective of the study should be more clearly defined (authors seems to only assess increasing prevalence of CKD or AKi according to the severity of COVID19).

Response: We have adjusted it as “to verify whether preexisting CKD is a risk factor for severe COVID-19, and whether AKI is a complication for severe COVID-19.”

3. Authors should provide SR registration number as in PROSPERO.

Response: We were trying to register this systematic review in PROSPERO. However, we gave up because of time urgency, and we published it in medRxiv in May 02, 2020 to attract more attention for the potential kindey injury. Therefore, we missed the opportunity to register it in PROSPERO.

4. In selecting the studies, AKI or CKD should be clearly defined, which seems vague in this study.

Response: AKI was clearly defined in six included studies. In five included studies, AKI was defined according to the Kidney Disease: Improving Global Outcomes (KDIGO) criteria. In addition, AKI was defined as any of renal functional indices beyond normal range in one included study. Unfortunately, none of the included studies mentioned the CKD diagnostic criteria probably because CKD is a preexisting disease. AKI is a complication of COVID-19, which probably cause most of the included studies did not mention its diagnostic criteria. Considering these reasons, AKI and CKD were not clearly defined in this study.

5. Studies selected are only from China, but those from other area should be included for applicability of the study results.

Response: As indicated in inclusion criteria, the publcation area and language were not limited in this systematic review. We were planning to include studies from all over the world to increase the credibility of results. However, all the included studies are only from China, which restrict its applicability in other areas.

6. Authors should provide at least one literature search formula, so that readers/reviewers can reproduce the study results.

Response: The literature search formula were as follws:

For English databases:

#1“SARS-CoV-2” (all fields or all text)

#2 “2019-nCoV” (all fields or all text)

#3 “COVID-19” (all fields or all text)

#4“novel coronavirus pneumonia” (all fields or all text)

#5 (#1 OR #2 OR #3 OR #4)

#6 “kidney” (all fields or all text)

#7 “renal” (all fields or all text)

#8 (#6 OR #7)

#9 (#5 AND #8)

For Chinese databases:

#1 “xin xing guan zhuang bing du” (subject)

#2 “xin xing guan Zhuang bing du fei yan” (subject)

#3 (#1 OR #2)

7. Risk of Bias table should be presented. For NOS score, only the total score is not enough but needs to present which item acquired the score in each selected study.

Response: With your suggestion, the detailed NOS score have be provided in the supplementary materials.

8. Last paragraph of the Discussion should not be the recommendation. Recommendation should be derived from the current study, but it did not provide any information for the recommended items.

Response: According to your suggestion, we have deleted the recommendation part.

9. Conclusion should be confined to the one derived from the results gained in this study. Need for CRRT cannot be concluded from the study.

Response: The result showed that severe group had higher ratio of receiving CRRT than nonsevere group (P < 0.00001, OR = 23.63, 95% CI: 8.34 to 66.95). It is concluded that CRRT is applied more frequently in the severe group than that in the nonsevere group. However, need for CRRT were not concluded from the study. We speculate that there is some kind of misunderstanding.

Reviewer #2: This review article evaluated involvement of CKD and AKI in COVID-19 patients. Due to the high impact of COVID-19 on public healthcare all over the world, I believe there is useful information in this manuscript. However, there are many inaccurate wordings in the current manuscript, which make it difficult to understand the appropriate conclusions from this study. Overall, I believe what the authors found were 1. CKD is a risk factor of severe COVID-19, and 2. Higher complication rates of AKI were found in severe COVID-19 patients when compared to non-severe cases. This is something already reported, but additional publications should be welcomed by researchers. Followings are my major comments.

1. The definition of severe or non-severe COVID-19 cases is unclear. Because of this, it was not possible to evaluate the manuscript appropriately. Does the severe case mean patients who required RRT, intubation, or something else?

Response: With in-depth understanding of COVID-19, the severity classification for COVID-19 is also changing according to China national diagnosis and treatment guidelines. In some earlier studies, patients were divided into two groups: severe group and nonsevere group. Subsequently, patients were divided into multiple groups, including mild, moderate, severe, and critical in later studies. Therefore, we divided all the included studies into two subgroups: severe group compared with nonsevere group, critical group compared with severe group. Severe COVID-19 is defined as respiratory rate ≥ 30 times/min, or oxygen saturation ≤ 93 % during resting state, or PaO2 / FiO2 ≤ 300 mmhg, or progress in imaging of lung lesions > 50%.

2. In the introduction, the number of COVID patient is outdated. Please update.

Response: With your suggestion, we have updated the number of COVID-19 patient.

3. At the end of introduction, authors said “to verify whether kidney diseases are involved in COVID-19”. This is an unclear statement. If the purpose of this study is to determine whether preexisting CKD is a risk factor for severe COVID19, then authors should explain so. If the purpose was to evaluate whether AKI is a complication in severe COVID19 cases, that should be also explained.

Response: Thank you very much for your kind reminder. Under your suggestion, we have revised the last sentence of the Introduction.

4. In study description, authors explained “All the included studies were performed in China”. What is the reason why studies from outside China were excluded?

Response: We were planning to include enough studies all over the world to lower region bias and increase the quality of evidence. However, the studies from outside China were excluded because of unavailable outcomes.

5. Due to unclear wording, it is difficult to understand the results. For example, in Comorbidity of CKD, authors said “the comorbidity of CKD was significantly increased in the severe group than that in the nonsevere groups.”. This sounds like the CKD was developed more frequently in severe group after COVID19 infection. But, I believe what authors meant was the patients with CKD were more likely to develop severe COVID19. The same is true to serum creatinine, blood urea nitrogen, and etc.

Response: Thank for your understanding. We have adjusted it as “propotion of preexisting CKD was significantly increased in the severe group than that in the nonsevere group”. In terms of serum creatinine, blood urea nitrogen and other outcomes, they are not preexisted.

6. In the result section, “serum creatinine” and “blood urea nitrogen” were repeated twice. This is confusing and needs to be corrected.

Response: “Serum creatinine” and “blood urea nitrogen” were compared in two differet subgroups: severe group compared with nonsevere group, critical group compared with severe group. In each subgroup, “serum creatinine” and “blood urea nitrogen” refer to specific values. Meanwhile, “abnormal serum creatinine” and “abnormal blood urea nitrogen” refer to the number of patients with elevated serum creatinine or blood urea nitrogen.

7. In discussion, authors claimed this is the first research to evaluate kidney involvement in COVID-19 patients. There are many papers published. This is probably overclaim and needs to be revised.

Response: We published the article in medRxiv in May 02, 2020. Afterwards, many similar papers published. Anyway, we are glad to revise the sentence.

---

## [Decision Letter · Decision Letter 1]

17 Dec 2020

The Chronic Kidney Disease and Acute Kidney Injury Involvement in COVID-19 Pandemic: A Systematic Review and Meta-analysis

PONE-D-20-22031R1

Dear Dr. Liu,

We’re pleased to inform you that your manuscript has been judged scientifically suitable for publication and will be formally accepted for publication once it meets all outstanding technical requirements.

Kind regards,

Tatsuo Shimosawa, M.D., Ph.D.

Academic Editor

PLOS ONE

Additional Editor Comments (optional):

One reviewer questioned why you do not register this study on PROSPERO. It is not mandatory and not our policy that metaanalysis is preregistered on PROSPERO but I encourage you to register to keep the transparency of the study.

Reviewers' comments:

Reviewer's Responses to Questions

**Comments to the Author**

1. If the authors have adequately addressed your comments raised in a previous round of review and you feel that this manuscript is now acceptable for publication, you may indicate that here to bypass the “Comments to the Author” section, enter your conflict of interest statement in the “Confidential to Editor” section, and submit your "Accept" recommendation.

Reviewer #1: All comments have been addressed

Reviewer #2: All comments have been addressed

2. Is the manuscript technically sound, and do the data support the conclusions?

Reviewer #1: Yes

Reviewer #2: Yes

3. Has the statistical analysis been performed appropriately and rigorously? 

Reviewer #1: I Don't Know

Reviewer #2: Yes

4. Have the authors made all data underlying the findings in their manuscript fully available?

Reviewer #1: Yes

Reviewer #2: Yes

5. Is the manuscript presented in an intelligible fashion and written in standard English?

Reviewer #1: Yes

Reviewer #2: No

6. Review Comments to the Author

Reviewer #1: The comments are almost satisfactorily answered except for the registration for PROSPERO. Since authors claim that this article was published in medRxiv on May 2nd, they should have enough time to register for PROSPERO. I would leave the decision to editors if SR/meta article could be published in PLOS ONE without registration to PROSPERO.

Reviewer #2: Authors addressed my concerns as much as possible by themselves. From now, I leave it to the editors to edit the wording to be accurate and more understandable.

7. PLOS authors have the option to publish the peer review history of their article (what does this mean?). If published, this will include your full peer review and any attached files.

Reviewer #1: No

Reviewer #2: **Yes: **Ryuji Morizane

---

## [Editor Report · Acceptance letter]

22 Dec 2020

PONE-D-20-22031R1 

The Chronic Kidney Disease and Acute Kidney Injury Involvement in COVID-19 Pandemic: A Systematic Review and Meta-analysis 

Dear Dr. Liu:

I'm pleased to inform you that your manuscript has been deemed suitable for publication in PLOS ONE. Congratulations! Your manuscript is now with our production department. 

Kind regards, 

on behalf of

Prof. Tatsuo Shimosawa 

Academic Editor

PLOS ONE